# Effects of replication domains on genome-wide UV-induced DNA damage and repair

**Yanchao Huang[1]ᵒ, Cem Azgari[2]ᵒ, Mengdie Yin[1], Yi-Ying Chiou[3], Laura A. Lindsey-Boltz[4], Aziz Sancar[4], Jinchuan Hu[1]\*, Ogun Adebali[2,5]\***

1 Shanghai Fifth People's Hospital, Fudan University, and Shanghai Key Laboratory of Medical Epigenetics, International Co-laboratory of Medical Epigenetics and Metabolism (Ministry of Science and Technology), Institutes of Biomedical Sciences, Fudan University, Shanghai, China, 2 Faculty of Engineering and Natural Sciences, Sabanci University, Istanbul, Turkey, 3 Graduate Institute of Biochemistry, National Chung Hsing University, Taichung City, Taiwan, 4 Department of Biochemistry and Biophysics, University of North Carolina at Chapel Hill School of Medicine, Chapel Hill, North Carolina, United States of America, 5 TÜBiTAK Research Institute for Fundamental Sciences, Gebze, Turkey

ᵒ These authors contributed equally to this work.
\* hujinchuan@fudan.edu.cn (JH); oadebali@sabanciuniv.edu (OA)

## Abstract

Nucleotide excision repair is the primary repair mechanism that removes UV-induced DNA lesions in placentals. Unrepaired UV-induced lesions could result in mutations during DNA replication. Although the mutagenesis of pyrimidine dimers is reasonably well understood, the direct effects of replication fork progression on nucleotide excision repair are yet to be clarified. Here, we applied Damage-seq and XR-seq techniques and generated replication maps in synchronized UV-treated HeLa cells. The results suggest that ongoing replication stimulates local repair in both early and late replication domains. Additionally, it was revealed that lesions on lagging strand templates are repaired slower in late replication domains, which is probably due to the imbalanced sequence context. Asymmetric relative repair is in line with the strand bias of melanoma mutations, suggesting a role of exogenous damage, repair, and replication in mutational strand asymmetry.

## Author summary

UV-induced damage can cause mutations during DNA replication, and crosstalk between replication and repair may influence mutagenesis. By integrating genome-wide damage, repair, and replication maps, we investigated the effects of replication domains on nucleotide excision repair. Early replication domains are repaired faster due to open chromatin; thus, they harbor fewer mutations. In addition, repair levels show strand asymmetry in the initiation zones of late replication domains, favoring leading strands independent of the ongoing replication. However, initiation zones of late replication domains have high AT content and exhibit a strong strand asymmetry with more T-tracts on lagging strands. Therefore, this biased repair, which coincides with melanoma mutational asymmetry, could be caused by a sequence content that leads to higher CPD formation and reduced

**Data Availability Statement:** Data sets produced in this study have been deposited to Sequence Read Archive with the bioproject id PRJNA608124. Pre-analyses codes of Damage-seq and XR-seq are

publicly available at https://github.com/
CompGenomeLab/xr-ds-seq-snakemake and all
the other codes are available at https://github.com/
CompGenomeLab/replicationRepair.

**Funding:** This study is supported by the National
Natural Science Foundation of China (NSFC)
[31870804], Program for Professor of Special
Appointment (Eastern Scholar) at Shanghai
Institutions of Higher Learning [2019], Shanghai
Outstanding Young Talent Program [2019],
innovative research team of high-level local
university in Shanghai, Shanghai Municipal Natural
Science Foundation [22ZR1413900] to J.H.; the
Scientific and Technological Research Council of
Turkey (TÜBİTAK) [118C320 to O.A.]. O. A. is
supported by EMBO Installation Grant [no: 4163]
and Science Academy, Turkey. The funders had no
role in study design, data collection and analysis,
decision to publish, or preparation of the
manuscript.

**Competing interests:** The authors have declared
that no competing interests exist.

repair. Our findings suggest that asymmetric damage formation and repair contribute to
mutagenesis asymmetry around replication initiation zones.

## Introduction

Nucleotide excision repair is a versatile mechanism responsible for removing bulky adducts
induced by various agents including UV irradiation, environmental pollutants like benzopy-
rene and chemotherapy drugs such as cisplatin [1]. Unrepaired damage interferes with DNA
replication and transcription, thus induces mutation and cell death, and finally results in can-
cer and aging [2–4]. Based on the mode of damage recognition, nucleotide excision repair can
be divided into transcription-coupled repair that initiates repair by stalled RNA polymerase II,
and global genome repair that recognizes double-strand distortion caused by the lesion [5]. In
both sub-pathways, following damage recognition, general repair factors including TFIIH,
XPA, RPA, and two endonucleases, XPG and XPF, are recruited to damage sites to make dual
incisions bracketing DNA adducts, releasing single-stranded fragments containing damage.
The resulting gaps are sealed by DNA polymerase and ligase to complete repair [6, 7].

Nucleotide excision repair crosstalks with chromatin factors and biochemical processes
including DNA replication and transcription. For instance, bulky adducts can block RNA
polymerase II and the stalled polymerase can act as a strong signal to initiate transcription cou-
pled repair, leading to preferred repair in the template strand of transcribed regions [8]. In
order to study the genome-wide regulation of nucleotide excision repair, sequencing methods
that measure bulky adducts and their repair were developed (reviewed in [9]). Among these
methods, Damage-seq can be used to detect the distribution of DNA damage at single nucleo-
tide resolution [10, 11], while XR-seq directly maps nucleotide excision repair, also at nucleo-
tide resolution, by capturing and sequencing the excised DNA fragments [12, 13]. These
methods, together with other similar methods, revealed that global genome repair is also regu-
lated by many factors including transcription, DNA accessibility, transcription factor binding,
nucleosome distribution and histone modification [6, 10, 14]. The heterogeneous repair pat-
terns are consistent with mutation distribution in skin cancers that are mainly caused by UV
exposure [15–19].

DNA replication is the key process for the fidelity of genetic information, and replication
error is the main source of mutations in many types of cancers [20]. Moreover, DNA lesions
induced by UV or other carcinogens cause point mutations when replicated, as mismatched
nucleotides are inserted during translesion synthesis by low-fidelity DNA polymerases. There-
fore, the crosstalk between DNA replication and repair is important for genome integrity.
However, the direct impact of replication on nearby repair remains to be elucidated. Unlike
transcription and chromatin states whose patterns are relatively constant in unchallenged
cells, DNA replication is highly dynamic and spreads throughout the whole genome. The
human genome possesses thousands of potential origins of replication (ORIs), but only a sub-
set of these origins is fired in a single cell during S phase. The order of firing is not predefined,
but ORIs in specific regions tend to be fired early or late. Early replication domains (ERDs) are
found in open chromatin and have fewer mutations in cancer genomes, whereas late replica-
tion domains (LRDs) overlap with compact chromatin and have elevated mutation frequency
[17, 21–24]. Moreover, because eukaryotic DNA replication is semi-discontinuous, mutation
distribution is also asymmetric as lagging strands have more mutations than leading strands
[25, 26]. Genomics studies also revealed replication related heterogeneity of mutation distribu-
tion in skin cancers [21, 27, 28], however, the genome-wide effects of DNA replication on

nucleotide excision repair as well as the contribution of repair as a function of replication to the mutation distribution is still unclear.

In this study, we performed EdU-seq, XR-seq and Damage-seq in UV-irradiated Hela cells which were synchronized at either early or late S phase. Repair and damage events were analyzed with respect to DNA strands and replication domains. Our results suggested that nucleotide excision repair was always faster in ERDs. Nevertheless, ongoing replication forks could exert an additional impact of promoting local repair in both ERDs and LRDs. In addition, repair efficiency exhibited a strand asymmetry favoring the leading template strands around initiation zones in LRDs, in line with the reduced mutation frequency of leading strands in melanoma.

## Results

### Genome-wide mapping of UV-induced damages and their repair in synchronized cells

In order to obtain cells replicating the same regions, HeLa cells were synchronized to G1/S by double thymidine treatment and released into S phase for different times. Flow cytometry results showed that at 2- or 4-hour time points, the majority of cells were in early or late S phase (S1A Fig), thus ERDs and LRDs should be the dominant replicating regions, respectively. To determine the replication domains, we performed EdU incorporation and sequencing (EdU-seq) at corresponding time points after releasing into S phase and obtained replicating regions in early and late S phases, which was in correlation with S1/S2 and S3/S4 phases of Repli-seq, (S2A Fig) respectively [29, 30]. The resulting ERDs and LRDs were similar as those determined by Repli-seq in asynchronized cells (S2B Fig).

To analyze damage and repair along with replication states, cells in either early or late S phase were irradiated with UV. Genomic distribution of both UV-induced damages, pyrimidine-pyrimidone (6–4) photoproducts [(6–4)PPs] and cyclobutane pyrimidine dimers (CPDs), were measured by Damage-seq immediately after UV, whereas the repair events were identified by XR-seq 12 min after UV in order to allow enough time for repair but short enough time that the replication status was unlikely to be affected (Fig 1A).

In agreement with the biochemical experiments as well as sequencing data from other cell lines [13, 31, 32], XR-seq from HeLa cells showed the excised oligomers were 22–30 nucleotides in length, with a median of 26 nucleotides (Fig 1B and S3B Fig). Dinucleotide distribution showed enrichment of TC [(6–4)PPs] or TT (CPDs) at predicted damage sites which were 5 to 6 nucleotides from the 3' end (Fig 1C and S3C Fig, Left). The specificity of Damage-seq was verified by enrichment of dipyrimidine at expected damage sites (Fig 1C and S3C Fig, Right), similar to previous results [10]. To eliminate the potential bias in damage formation caused by sequence context [10] around TSS and TES regions, we initially normalized repair events (XR-seq) by the damage quantities (Damage-seq) for each genomic window (see Methods), which we referred to as "repair rate" throughout the paper. There was no difference of repair rate between transcription template and non-template strands for both damages (Fig 1D), suggesting that transcription coupled repair had no apparent contribution at this early 12 minute time point. Lack of transcription coupled repair at 12 minutes was also observed in NHF1 cells in a recent study [33], indicating that this phenomenon is not specific to HeLa cells. Moreover, preferential repair of template strands was seen at 2 hours after UV (S3D Fig), indicating transcription coupled repair is not defective in HeLa cells. Therefore, we could clearly assess the association between global repair and replication at 12 min, without the need for considering the effect of transcription coupled repair.

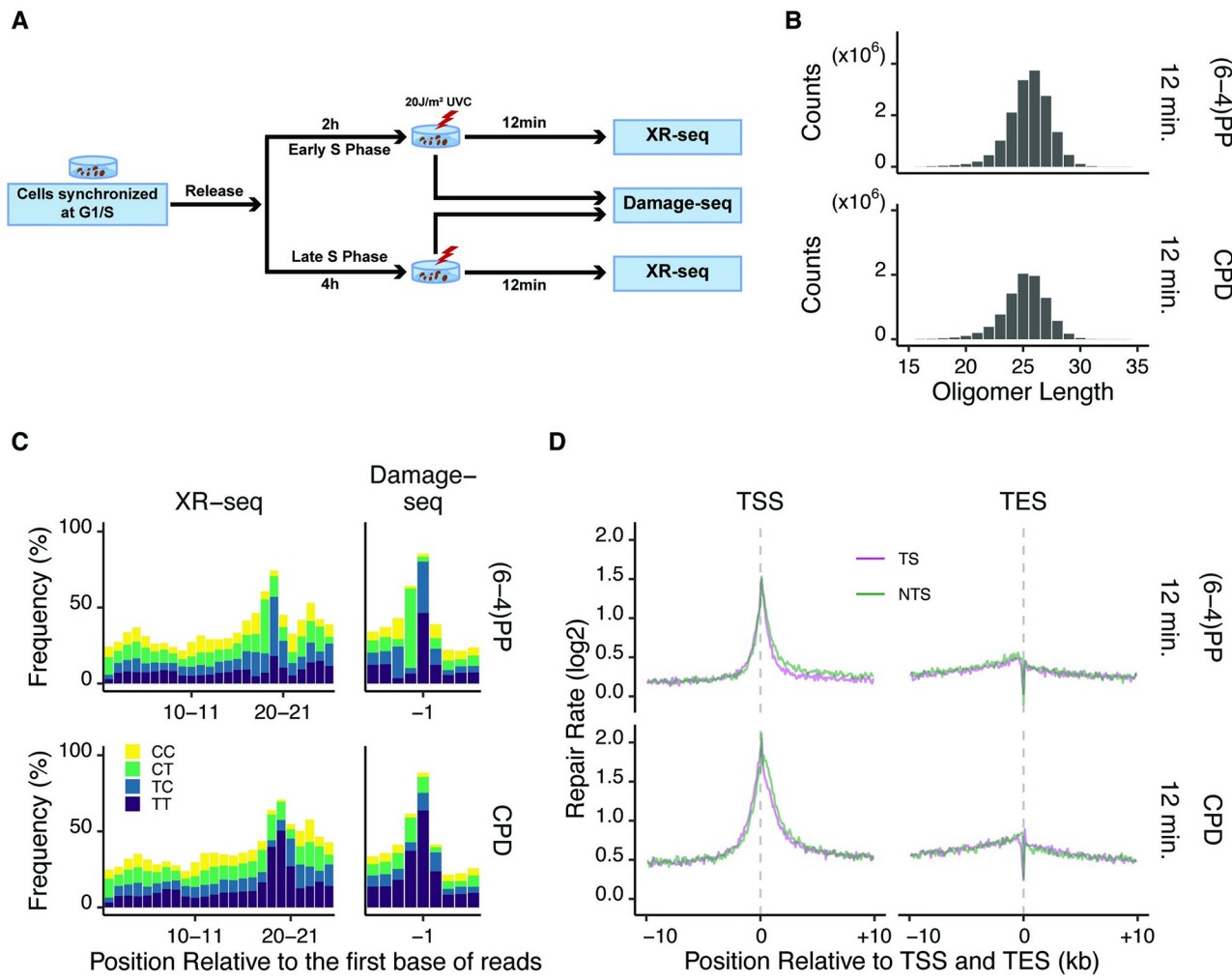

**Fig 1. Damage and repair maps of UV-irradiated synchronized HeLa cells.** (A) Schematic of experiment design. HeLa cells were synchronized and exposed to 20J/m² UVC. Immediately or 12 minutes after UV irradiation, they were collected to perform Damage-seq or XR-seq, respectively. (B) Distribution of oligomer length of XR-seq reads of (6–4)PPs and CPDs in late S phase. (C) Dipyrimidine content of 26-nucleotide-long XR-seq and 10 nucleotides around the first nucleotide of Damage-seq reads of (6–4)PPs and CPDs in late S phase. Because the DNA polymerase stops right before adducts during Damage-seq, the expected damage sites should be at -1 position. (D) Repair rates [log2 (XR-seq/Damage-seq)] of template strands (TS) and non-template strands (NTS) around transcription start sites (TSS) and end sites (TES). (B-C) Replicate A is shown. (D) Replicate A and B are combined.

## Ongoing replication fork stimulates nearby nucleotide excision repair

Damage and repair data were mapped to the replication domains determined by EdU-seq. Although ERDs had, in general, more damage and repair reads than LRDs (S4 Fig), partially because ERDs were replicated earlier and had more total DNA in both early and late S phase cells, this effect was eliminated by the repair rate calculation. To further reduce the bias of sequence context, repair rates were normalized with simulated damage and repair signals that were produced by profiling sequence frequencies to obtain regions with similar profiles from input sequencing data. For both early and late S phases, normalized repair rates of CPDs peaked at the center of ERDs, while LRDs exhibited an opposite pattern (Fig 2A), presumably because ERDs are correlated with open chromatin regions and more accessible to repair proteins, and vice versa for LRDs [22, 29]. However, this trend is absent (in ERDs) or less apparent

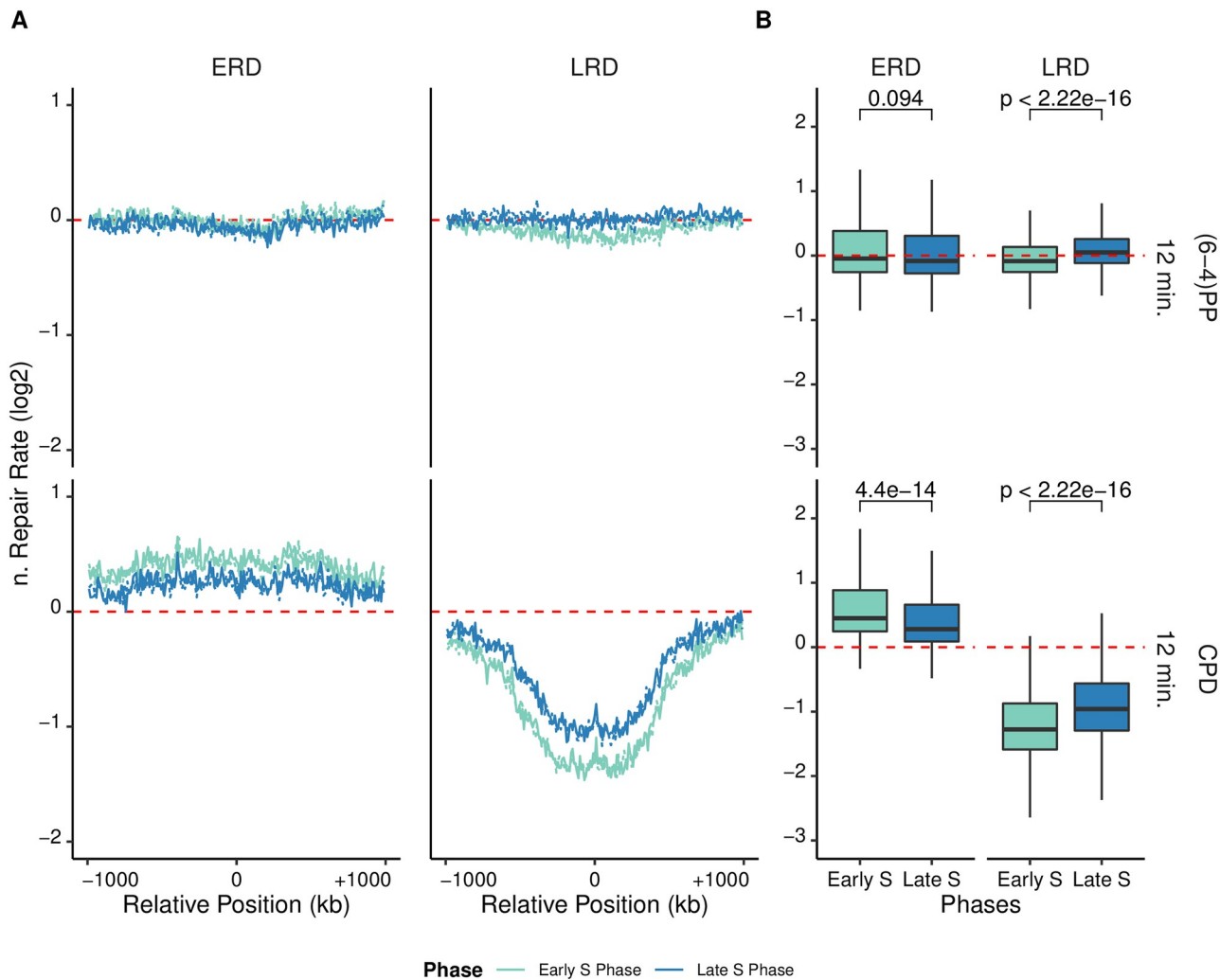

**Fig 2. The shift of repair rates with replication timing.** (A) normalized Repair rates [log2 (XR-seq$_{real/simulation}$/Damage-seq$_{real/simulation}$)] of (6–4)PPs (Top) and CPDs (Bottom) around the center of ERDs (left, n = 118) and LRDs (right, n = 237). Dashed lines were the plus strands and straight lines were the minus strands. (B) Boxplot of normalized repair rates of (6–4)PPs and CPDs for both ERDs (left, n = 266) and LRDs (right, n = 541). The wilcoxon test was used to assess the significance of differences between early and late S phases. Replicate A and B are combined.

(in LRDs) for (6–4)PP, in line with the fact that the repair of (6–4)PP is less influenced by chromatin organization [10, 17].

Repair rates in both ERDs and LRDs changed as the cell cycle moved from early to late S phase. Because ERDs and LRDs were being replicated in early and late S phases, respectively, the differences of repair rate between two phases reflected the impact of ongoing replication on nearby repair. As shown in Fig 2, ERDs were repaired faster in early S phase and LRDs were repaired faster in late S phase, indicating that repair rate was elevated when the region started to be replicated (log2FC between early and late S phase: (6–4)PPs in ERDs 0.05; (6–4)PPs in LRDs -0.13; CPDs in ERDs 0.2; CPDs in LRDs -0.3). Moreover, we plotted chromosome-wide replication timing, repair rates, and fold change of repair between early and late S phases in chromosomes 1 and 19 (S5 Fig). The results also suggested that while ERDs always have higher repair rate than LRDs, both of them are repaired faster when being replicated in general. It was reported that transcription could also stimulate global genome repair

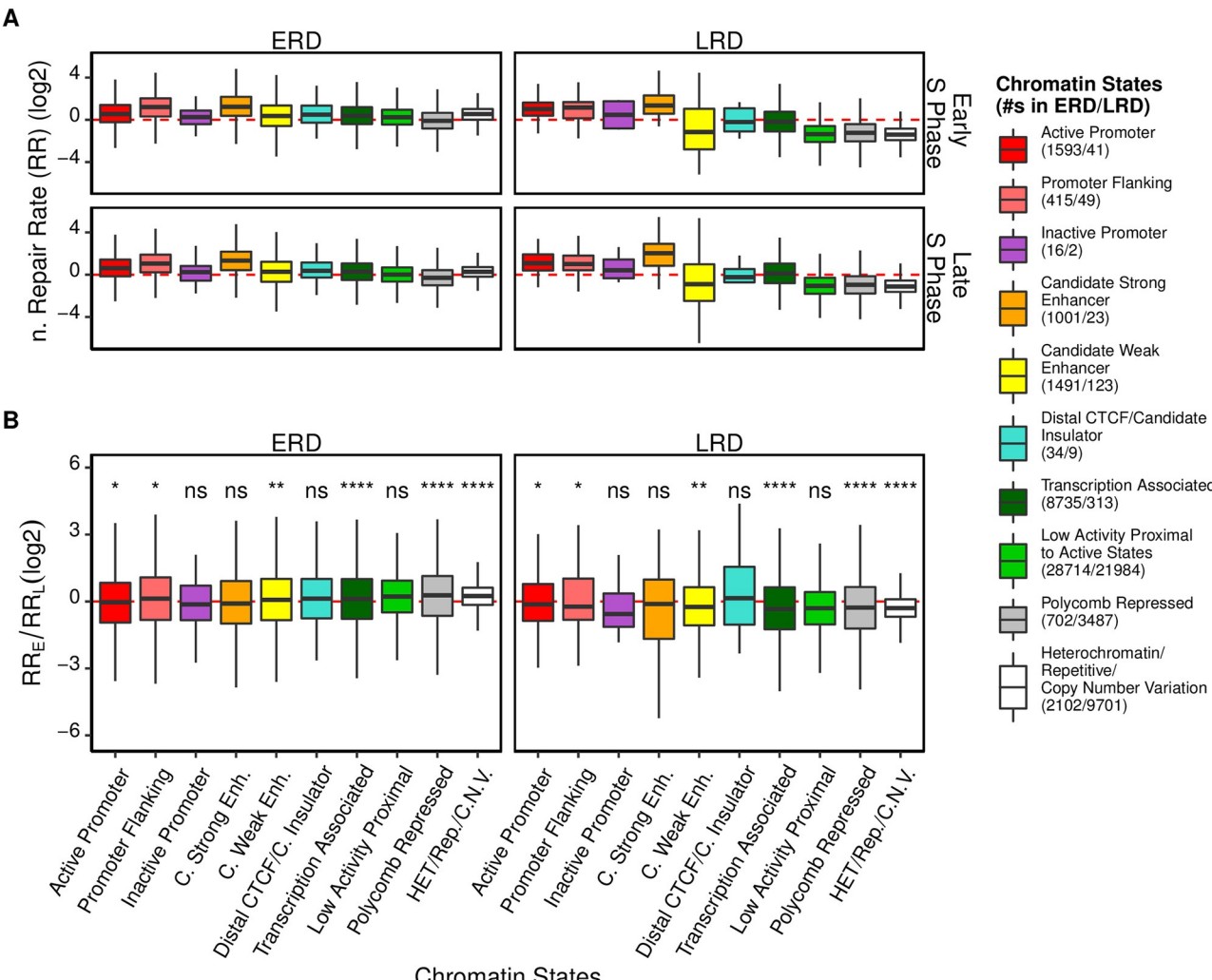

**Fig 3. The effect of chromatin states along with replication timing on repair.** (A) Boxplot of normalized repair rates [log2(XR-seq$_{real/simulation}$/Damage-seq$_{real/simulation}$)] of CPDs at 12 minutes after UV at early and late S phases for each chromatin state. (B) Relative difference of early S phase and late S phase repair rates [log2(RR$_{Early S Phase}$/RR$_{Late S Phase}$)] for each chromatin state. The wilcoxon test was used to assess the significance of differences between [log2(RR$_{Early S Phase}$/RR$_{Late S Phase}$)] and 0. Replicate A and B are combined.

independent of transcription coupled repair [28, 34]. Similar as the impacts of replication domains, the effects of replication phases were less prominent for (6–4)PPs than for CPDs, which could be explained by the fact that (6–4)PPs are more efficiently recognized by global genome repair system and less affected by chromatin states than CPDs [10, 17].

Because replication might affect repair through chromatin structure, it is intriguing how this effect differs in various chromatin states. In order to address this question, chromatin states of HeLa cells segmented by ChromHMM were retrieved from UCSC [35] and integrated with replication domains. Relative repair rates of ERDs or LRDs in each state from either early or late S phase cells were calculated (Fig 3 and S6 Fig). Consistent with previous reports [11, 17], active chromatin states in both ERDs and LRDs were repaired efficiently. To assess the influence of ongoing replication, repair rates of these segments in early and late S phases were compared (Fig 3 and S6 and S7 Figs). As mentioned above, the repair of (6–4)PPs was less affected by the chromatin states as well as the replication phases (S6 Fig). For this reason, we

focused on CPD repair. As shown in Fig 3, in general, the difference between two phases is minor and/or less significant in active chromatin states (e.g., strong enhancers) which were repaired more efficiently. In contrast, the repair rates of "Transcription Associated" and inactive regions like "Polycomb Repressed" and "Heterochromatin/Repetitive/Copy Number Variation" were elevated remarkably with the replication progression, i.e., the repair rates of these regions belonging to ERDs are significantly higher in early S phase than that in late S phase, and vice versa for the regions belong to LRDs. This effect is more apparent for LRDs than ERDs in "Transcription Associated", "Low Activity Proximal" and "Heterochromatin/Repetitive/Copy Number Variation" regions (S7 Fig). Moreover, the odds ratio of ERDs versus LRDs in inactive and transcription-associated regions supported this repair rate difference (S8 Fig). Therefore, replication preferably promotes CPD repair in inactive and transcription-associated regions. However, the result is complicated, e.g., some regions with low repair capacity like "low activity proximal" regions are not significantly affected by replication. Noteworthy, there are two factors affecting the change of repair between early and late phases in each state: to what extent replication relaxes local chromatin and how the change of chromatin compaction influence repair. Here we only considered the latter one, while the former one is unclear in human cells.

## Replication-related strand asymmetry of nucleotide excision repair

Eukaryotic DNA replication is an asymmetric process on two strands which results in a replication-related bias of mutations between leading and lagging strands. It is not clear whether nucleotide excision repair is influenced by the semi-discontinuous replication process thus contributing to the mutation bias. In order to make leading and lagging strand assignments, Okazaki fragment sequencing (OK-seq) data from HeLa cells was retrieved [36]. Although an individual ORI firing is a random event in a single cell, OK-seq revealed that there are sets of closely positioned ORIs, defined as "initiation zones". The leading and lagging strands switch around initiation zones, which allows the study of replication-associated strand asymmetry (Fig 4A).

In line with a previous analysis of CPD Damage-seq and XR-seq data from human skin fibroblasts [27], the lagging strands had both more damage and repair in HeLa cells (Fig 4B and S9 Fig). However, the simulated damage and repair signals showed a similar asymmetric pattern (Fig 4B and S9 Fig), suggesting that the lagging strand bias seen in both cell lines was likely due to the sequence context. Moreover, the asymmetry in LRDs was stronger than that of ERDs, suggesting that the sequence context bias is more apparent in LRDs. The same trend was also observed for (6–4)PP, albeit weaker than CPD (S10 Fig). In order to eliminate the influence of sequence context, normalized repair rate in ERDs and LRDs was calculated as described above by normalizing actual repair rate (the ratio of actual repair to damage) to simulated repair rate (the ratio of simulated repair to damage). Virtually no obvious difference between the two strands was observed for (6–4)PPs (S11 and S12 Figs), whereas a preferred repair on the leading strands was observed for CPDs in LRDs (Fig 4C).

To determine whether a longer repair time would affect the efficiency of leading versus lagging strand repair, synchronized HeLa cells were incubated for 2 h after UV irradiation (S3A Fig). Flow cytometry showed that replication progress was delayed but not stopped after UV (S1B Fig), and EdU-seq indicated that the organization of replication domains was not disrupted by UV treatment (S2B Fig). Because the majority of (6–4)PPs was removed after 2 h [31], Damage-seq and XR-seq were only performed for CPDs at this time point. As shown in Fig 4D and S13 Fig, leading strand repair was faster than lagging strand and the difference was more prominent in both ERDs and LRDs than 12 min, while LRDs had much stronger strand

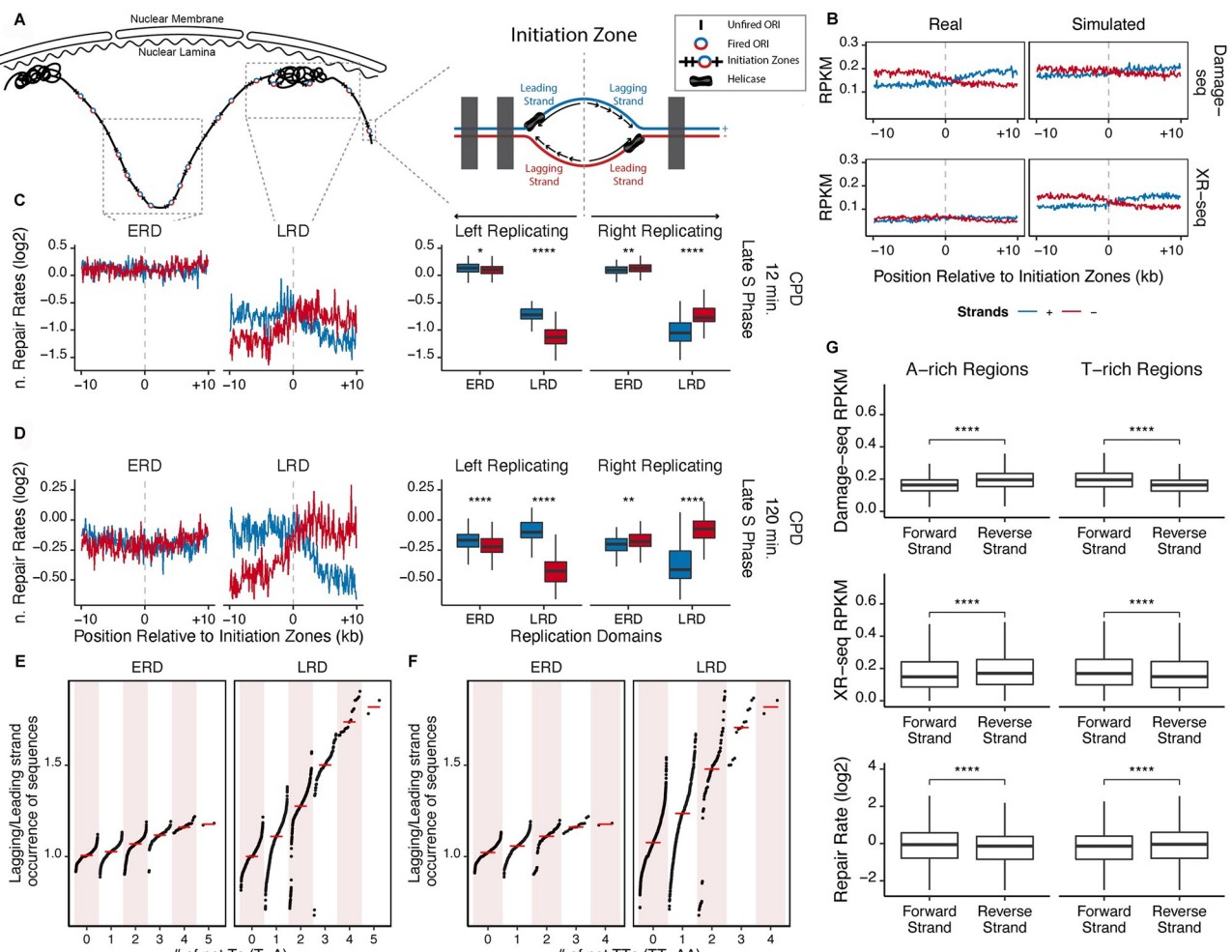

**Fig 4. Repair rate and sequence context asymmetry around initiation zones in ERDs and LRDs.** (A) Schematics of replication domains (Left), and initiation zones (Right). (B) Damage and repair profiles of real and simulated reads around initiation zones in LRDs for CPDs at 0 minutes (for damage), 12 minutes (for repair). (C-D) (Left) CPD normalized repair rates [log2 (XR seq_{real/simulation}/Damage-seq_{real/simulation})] at 12 minutes (C), and 2 hours (D) in late S phase around initiation zones that were separated into corresponding replication domains. (Right) The boxplot of windows in left replicating and right replicating directions are shown separately for plus and minus strands. Paired wilcoxon test was used to assess the significance of differences between the strands. Number of initiation zones in ERDs: 2130; in LRDs: 1450. (E-F) Correlation of relative occurrence of all mono- to penta-nucleotide sequences on the lagging and leading strand and the number of net Ts (T-A) (E) and net TTs (TT-AA) (F) for each sequence. Each dot represents a sequence (separate dots for left and right replicating directions), and the red bar represents the median of the Lagging/Leading strand values for each net T value. Replicate A and B are combined. (G) Strand differences of damage, repair, and repair rate in A-rich and T-rich regions. Boxplot of Damage-seq (top), XR-seq RPKM values (middle), and log2 transformed repair rates (bottom) of asynchronized CPDs at 12 minutes in A-rich and T-rich (see Methods).

bias than ERDs at 2 h. This conclusion was confirmed by the accumulated repair pattern calculated by subtracting damage at 2h from those at 0h [(Damage-seq_{0h} − Damage-seq_{2h})/ Damage-seq_{0h}] (S13 Fig). However, transcription-coupled repair might play a role at 2 h but not 12 min (Fig 1D and S3D Fig). Indeed, leading strands around initiation zones are transcribed more than the lagging strands due to the asymmetrical distribution of gene directions around the initiation zones (S16A Fig). Nevertheless, after excluding all the annotated transcribed regions, the trend did not change (S15 and S16B Figs).

It is worth noting that the above trend was observed in both early and late S phases when ERDs and LRDs were being replicated, respectively. This phenomenon ruled out the

possibility that the strand asymmetry of normalized repair rate around initiation zones is the direct result of ongoing replication. Therefore, we hypothesize that the intrinsic sequence feature around initiation zones in LRDs determine the strand bias of repair rate. Notably, a stronger strand asymmetry of the real CPDs than that of simulated ones were observed in LRDs at 0 min after UV irradiation, indicating the effect of local sequence context on asymmetric damage formation as well (Fig 4B and S9 Fig). Asymmetric composition of poly T-tracks is thought to be the determinant of initiation zones, while LRDs are reported to have more ATs than ERDs [37]. Thus, we compared the ratios of T-rich mono- to penta-nucleotide(s) on leading and lagging strands around the initiation zones in ERDs or LRDs to find out the most imbalanced local sequences (S17–S21 Figs). Intriguingly, the results suggested that ERDs/LRDs and leading/lagging strands have apparently different sequence features. Firstly, initiation zones in LRDs have a higher AT content than those in ERDs (62% vs 54%). Regarding the strand bias, although the lagging strands contained more Ts than the leading strands in both ERDs and LRDs, the damage asymmetry was more obvious in LRDs (S17 Fig). Analyses of the distributions of all mono- to penta-nucleotide(s) on two strands revealed that the extent of strand asymmetry is positively correlated with the value of "net T" (T-A) in a given sequence, and this trend is more apparent in LRDs (Fig 4E). Net TT (TT-AA) and to a lesser extent TC (TC-GA), but not net C or CC, showed the same pattern as net T (Fig 4F and S22–S24 Figs), indicating this pattern is specific to T-containing tracks. Notably, most CPDs are formed on T-containing dipyrimidines (TT, CT and TC), and the strand asymmetry of T-rich sequence (preferring lagging strand especially in LRDs) coincides with asymmetric damage formation and repair. Therefore, the intrinsic sequence feature might be responsible for the observed strand bias of CPD formation and repair. Consistent with this explanation, regions with strong A/T bias also have similar strand asymmetry of repair (Fig 4G).

## Replication-related strand asymmetry of mutation distribution in melanoma

UV-induced C to T mutation in dipyrimidines is the dominant mutation signature in skin cancer [38]. We questioned whether the asymmetric damage formation and repair of CPDs that we observed could impact mutagenesis during cancer development. Melanoma mutation data was retrieved from the International Cancer Genome Consortium (ICGC) data portal (see Methods). Because there was no available OK-seq data from melanoma cells, we generated initiation zones in GM06990 (lymphoblastoid) and IMR90 (normal lung fibroblast) cell lines [36, 39] (see Methods), in addition to initiation zones of HeLa. Then, we overlapped the initiation zones of these cell lines to obtain the common initiation zones that could also be potentially conserved in melanoma cells (Fig 5A). Furthermore, we calculated the initiation zone scores in HeLa cells (adjusted OK-seq slopes, see Methods) for each initiation zone and found that common initiation zones had significantly higher initiation zone scores than the unique ones (Fig 5B), which means high-scored initiation zones are likely to be conserved among cell lines. Then, we separated the common initiation zones into quartiles based on their initiation zone scores, then assigned the mutations in intergenic regions to each strand around initiation zones and calculated the absolute difference between strands for each quartile (Fig 5C). Remarkably, with increasing initiation zone scores, the asymmetry in mutation asymmetry drastically increased in LRDs but not in ERDs. Lastly, the mutation frequency of common initiation zones was normalized to the counts of corresponding dinucleotides (see Methods) to eliminate any bias caused by the nucleotide content and plotted separately for replication domains (Fig 5D). Intriguingly, mutation distribution regarding replication domains and strand asymmetry exhibited connection with damage distribution as well as repair profiles

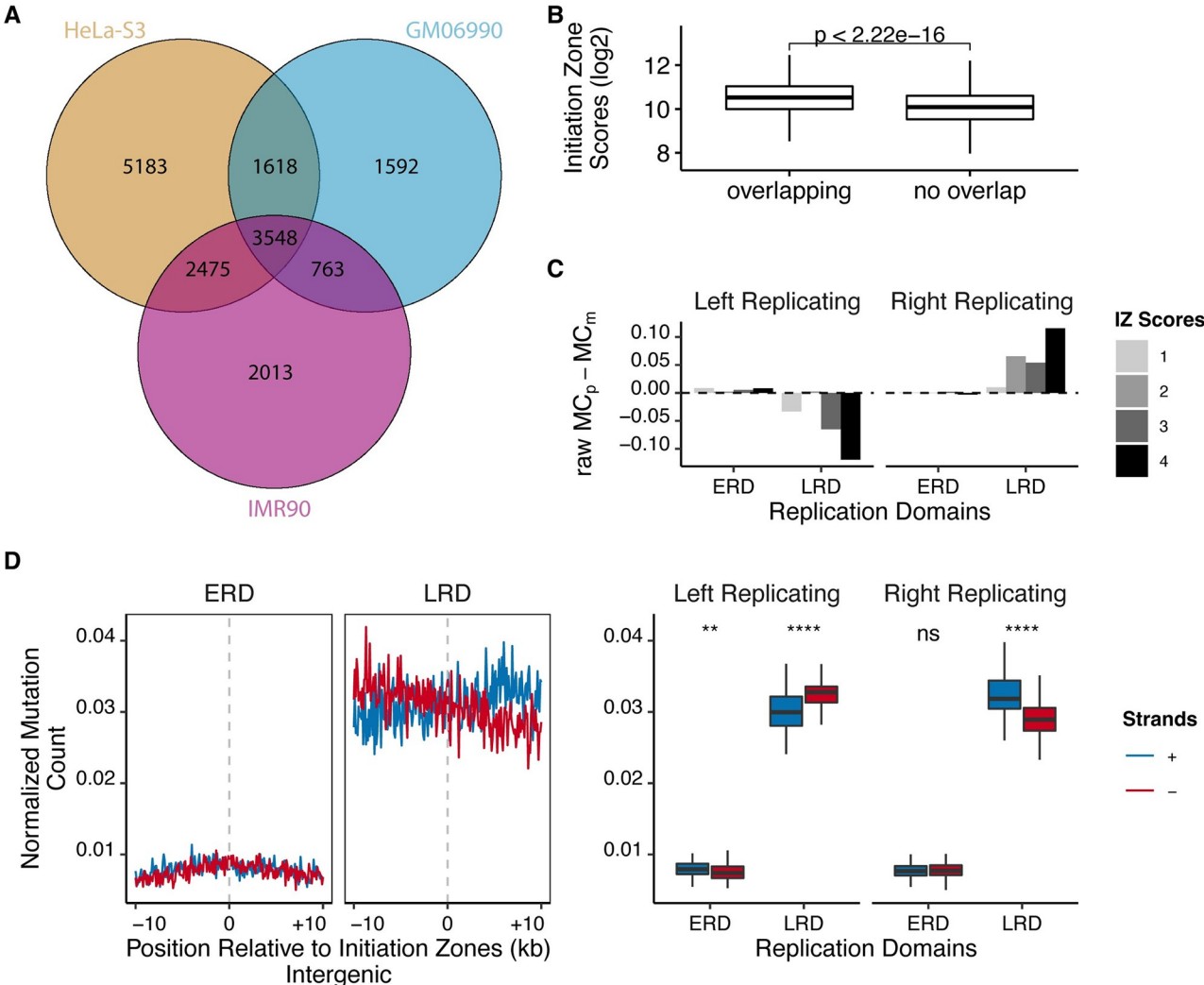

**Fig 5. Melanoma mutation asymmetry around common replication initiation zones.** (A) Venn diagram illustrating the shared initiation zones between HeLa, GM06990, and IMR90 cell lines. (B) Boxplots of HeLa initiation zones' scores that are common in all 3 cell lines (overlapping) and the ones that are unique to HeLa cells (no overlap). The wilcoxon test was used to assess the significance of difference. (C) Common initiation zones (20 kb long in 200 windows) were separated into quartiles based on their scores (OK-seq slope values) for each replication domain, and absolute mutation count difference between strands were calculated. Quartiles 1 and 4 correspond to the lowest and highest initiation zone scores, respectively. (D) (Left) Normalized TC>TT and CC>CT mutations (MC: mutation per [TC]C per region) around initiation zones that were separated into corresponding replication domains. (Right) The boxplot of windows in left and right replicating directions are shown separately for plus and minus strands. Paired wilcoxon test was used to assess the significance of differences between the strands. Only mutations in intergenic regions were considered. Raw MC: not normalized mutation counts; $MC_p$: mutation counts of plus strand; $MC_m$: mutation counts of minus strand.

(S25 and S15B Figs). Specifically, in agreement with previous studies [23, 24, 40], the mutation levels at LRDs were higher than ERDs, which was reversely correlated with the repair profile. Furthermore, mutation level showed strand asymmetry favoring lagging strands, and the difference was more obvious in LRDs than ERDs, coinciding with the expectation from the asymmetric damage formation and repair (S15B Fig and Fig 5D).

## Discussion

DNA adducts such as UV-induced pyrimidine dimers may cause base substitutions when they are replicated, as they can stop replicative DNA polymerases and stimulate the switch to

translesion DNA polymerases some of which incorporate incorrect bases [4]. If damage is repaired before replication, no mutation will be introduced. Nevertheless, there are several obstacles to investigating the effect of replication on repair. Firstly, damage-induced stalling of replicative DNA polymerases activates an intra-S phase checkpoint and delays replication [41]. Traditional assays measure DNA repair by comparing damage levels at different time points, however, replication does not proceed normally during this period. Secondly, the whole genome is copied once within the S phase which is usually 6–8 hours in human cells. Therefore, for any single cell, only a small fraction of the genome is being replicated, which means even if replication has a strong effect on local repair, it is hard to be detected by whole-genome-level analysis. Thus, the moderate decrease of repair during S phase which is greatly enhanced by ATR inhibition [42] seems to be the consequence of indirect effects.

Nonetheless, DNA replication may affect nearby repair in different ways. DNA adducts can block both replicative DNA polymerases and RNA polymerase II, and it is well known that the latter can trigger transcription coupled repair. In addition, transcription can relax local chromatin and enable the access of repair factors, thus facilitating global repair in an indirect way [34]. In the case of DNA replication, it has been reported that stalled replication forks can initiate the repair of inter-strand crosslink (ICL) [43]. Theoretically, nucleotide excision repair requires intact double-stranded DNA as a substrate [44], thus the single strand-double strand junctions generated by stalled DNA polymerases are not suitable substrates for repair. Some damages are repaired in a post-replicative way or even left unrepaired to the next cell cycle [45]. On the other hand, similar to transcription, DNA replication can also relax local chromatin [46, 47] and has the potential to directly promote repair. In order to distinguish the contribution of these two possible mechanisms, it is necessary to study the distribution of repair at replicating regions as we have performed here instead of measuring it as a whole.

## DNA replication promotes local repair by relaxing surrounding chromatin

In this study, genomic distributions of damage and repair were measured in cells synchronized in early or late S phase, in which ERDs or LRDs were the primary replicating regions, respectively. Not surprisingly, ERDs are repaired faster than LRDs in both early and late S phases because ERDs correspond to open chromatin in general and are more accessible to the repair machinery [29]. This result indicates that in addition to mismatch repair [21], repressed nucleotide excision repair also contributes to the elevated mutation frequency in LRDs in cancer genomes.

Despite the quantitative differences between the effect of replication on ERD and LRD repair, both ERDs and LRDs are repaired faster when they are being replicated, suggesting a positive impact of replication on local repair which is likely due to the effect of ongoing replication to relax surrounding chromatin (Fig 6A). This effect is not so evident for (6–4)PPs whose repair is much more efficient and less influenced by chromatin status [17].

When cells move from early S phase to late S phase, the repair of individual chromatin state changes in the same way as the total ERDs or LRDs, respectively. However, the difference between early and late S phases varies among chromatin states (Fig 3). Notably, the local chromatin structure would be disassembled before replication and restored post replication, and the dynamics of restoration might determine the effects of replication on local repair. A recent study in yeast indicated that the dynamics of chromatin restoration is not uniform throughout the genome [48]. For instance, the epigenetic markers of transcription (e.g. H3K36me3 and H3K79me3) were associated with slow chromatin restoration, consistent with our finding that transcribed regions have a remarkable change in CPD repair when replication forks move from ERDs to LRDs. Therefore, replication can stimulate surrounding repair probably by

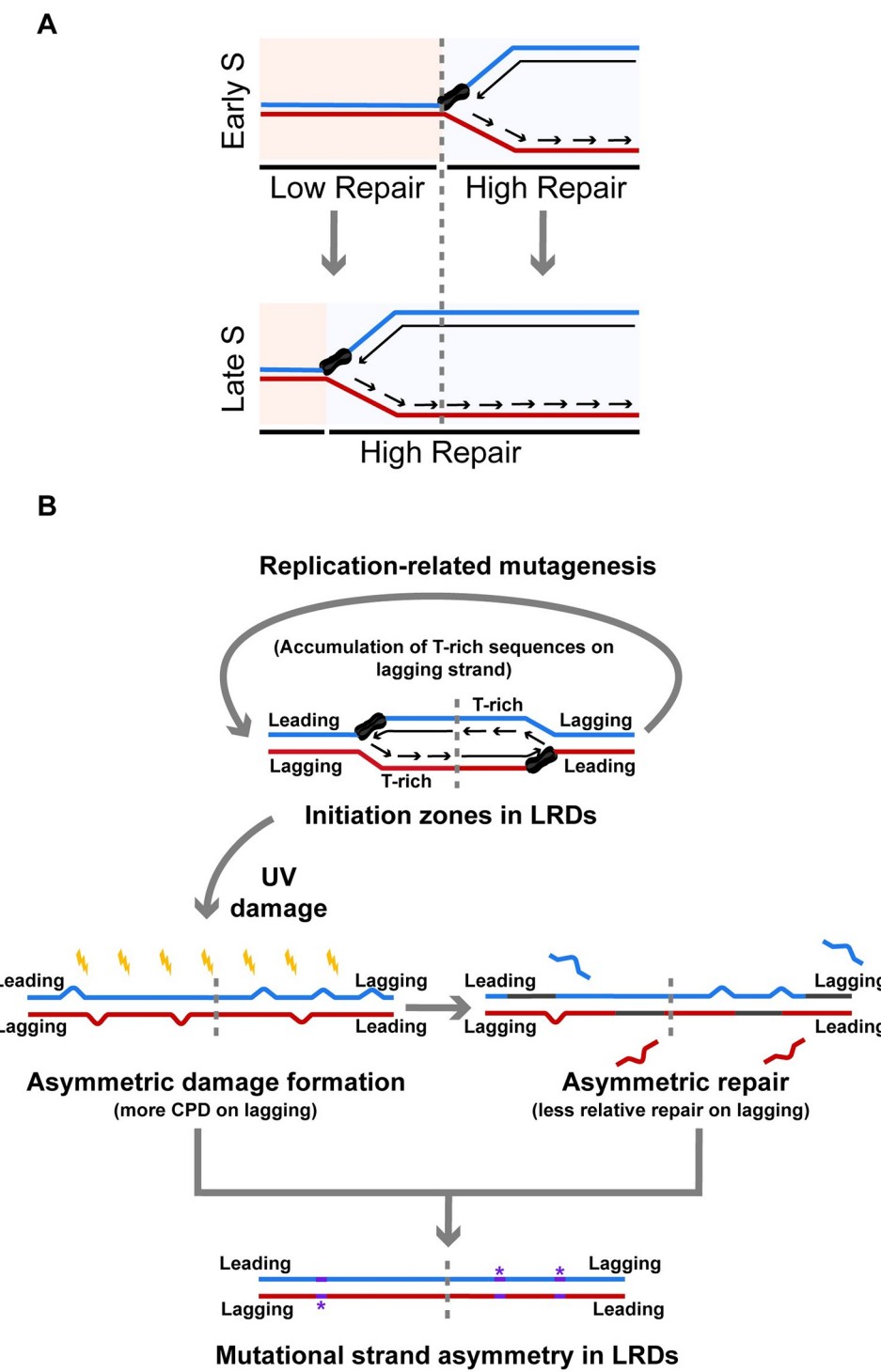

**Fig 6. The impacts of replication domains on damage formation and repair and related mutagenesis.** (A) Relaxation of compact chromatin by DNA replication progress promotes local repair. (B) Model showing the relationship among replication, sequence context, CPD formation, nucleotide excision repair and mutation accumulation in LRDs. Replication could cause the accumulation of T-rich sequences on lagging strands in LRDs, which contributes to favorable CPD formation and repressed repair and finally results in the strand asymmetry of mutation in the melanoma genome. Blue and red strands represent the plus- and minus-strands, respectively. Gray strands represent the resynthesized strands and C to T mutations are represented as purple stars.

relaxing local chromatin, especially for CPD whose repair is more dependent on chromatin states.

## Strand asymmetry of UV-induced DNA damage, repair, and mutagenesis in initiation zones of LRDs

Besides the discrepant mutation densities between ERDs and LRDs, replication-associated mutational strand asymmetry is widespread in multiple cancers [21, 25]. This phenomenon is usually attributed to replication-related processes such as POLE- and APOBEC-related mutagenesis. However, skin cancers, which are clearly related to UV-induced DNA damage, also show weak but significant replicational strand asymmetry besides the strong transcriptional strand asymmetry caused by transcription-coupled repair. Here we identified asymmetric UV-induced damage formation and relative repair rate in initiation zones. However, this asymmetry is much more apparent in LRDs than ERDs regardless of replication status.

Further analysis suggested two distinct sequence features around initiation zones. Firstly, LRDs have a higher AT content than ERDs. Secondly, there is a bias of the distributions of T-rich sequences such that the lagging strands have more T-rich sequences than leading strands. The extent of bias is positively correlated with the number of Ts and is much stronger in LRDs than in ERDs for a given T-rich sequence. Since TT is the major underlying sequence of CPD, the asymmetry of damage formation and repair is likely due to the imbalanced sequences. Notably, the strand asymmetry in actual Damage-seq is more substantial than that in simulated data, whereas actual XR-seq has weaker asymmetry than simulated data. In other words, the strand asymmetry of damage formation and repair does not simply reflect the biased sequence context. Instead, the sequence feature of leading and lagging strands exerts additional and opposite impacts on damage formation and repair. This observation is consistent with the fact that the discrepancy between leading and lagging strands is enlarged for both simulated and actual damage after 2 h repair. However, the mechanism for how the sequence influences damage formation and repair is unclear.

T-rich sequences on lagging strands in LRDs may exert contrary effects on CPD formation and repair, both of which should result in the elevation of mutation on lagging strands in LRDs as we observed in the melanoma genome. The asymmetric sequence feature in LRDs might come from undefined but clearly a replication-related mutation processes since replication-related mutagenesis always causes more mutations in LRDs as well as replicational strand asymmetry [21]. Notably, only C-containing CPDs (TC, CC, and CT) can cause the distinctive C to T mutation (signature 7, mainly TC to TT) [49], but the major CPD site identified by both Damage-seq and XR-seq is TT. However, as C/Gs are symmetrically distributed on both strands, the distribution of C-containing dipyrimidines (TC and CT) follows the same trend as TT (S18–S21 Figs), and the T-rich sequences should have similar effects on the formation and repair of C-containing CPDs as that on TT-CPD. In conclusion, although it cannot be ruled out that ongoing replication forks have a direct impact on the formation and repair of UV-induced damage, the strand asymmetry of damage formation and repair in LRDs is likely a result of imbalanced sequences in these regions. It is thus responsible for the mutational strand asymmetry in LRDs in the melanoma genome.

In summary, our results have shown that replication domains can affect the formation and repair of UV-induced damage and related mutagenesis in different ways. Firstly, ERDs are always repaired faster than LRDs since genome-wide repair profile mainly depends on chromatin compaction and transcription activity and ERDs are correlated with active regions, while ongoing replication can exert an additional repair enhancement probably by relaxing local chromatin (Fig 6A). Such an enhancing effect can potentially reduce damage-induced

mutations. On the other hand, during evolution, replication requirements and constraints can drive the accumulation of T-tracks on lagging strands over leading strands as well as influence the total AT content in initiation zones of LRDs by uncertain processes. Our damage and repair maps show that those T-rich sequences exhibit elevated damage levels and reduced repair rates independent of ongoing replication. Further analysis of UV-related melanoma mutations confirms the correlation between mutational strand asymmetry and the damage and repair bias in LRDs (Fig 6B). Mutation distribution in cancer genomes provides important information about the origin and development of cancer. Our findings reveal previously unknown effects of damage formation and repair on mutational strand asymmetry in specific replication domains of cancer genome.

## Materials and methods

### Cell culture and treatments

HeLa S3 cells (purchased from ATCC) were cultured in DMEM medium supplemented with 10% FBS and 1% penicillin/streptomycin at 37˚C in a 5% atmosphere CO2 humidified chamber. Cells were synchronized at late G1 phase by double-thymidine treatment, then released into S phase by removing thymidine. In brief, for first thymidine treatment, thymidine was added to cells at 50% confluence to a final concentration of 2 mM. After 18 hours, cells were released by washing with PBS and cultured in fresh medium for 9 hours. Then cells were treated with 2 mM thymidine for 15 hours and released into S phase for indicated time before UV irradiation. Cells were irradiated with 20 J/m$^2$ of UVC, then collected immediately or after incubation at 37˚C for indicated time for the following assays. All experiments were performed in duplicate. Replicate A and B were combined throughout the analyses.

### Flow cytometry analysis

Cells were trypsinized, Cells were trypsinized, PBS washed and fixed in 70% (v/v) ethanol at −20˚C for at least 2 hours, then stained in the staining solution (0.1% (v/v) Triton X-100, 0.2 mg/ml RNase A and 20 μg/ml propidium iodide in PBS) for 30 min at room temperature. Then S phase progression of the cells was analyzed by a flow cytometer (Beckman Coulter CyAn or Becton Dickinson FACSCanto II).

### EdU incorporation and sequencing (EdU-seq)

To determine the replicating regions in either early or late S phase, HeLa cells in early or late S phase (released into S phase for 2 and 4 h, respectively) were incubated with 10uM EdU (Sigma) for 15 min before collection. To determine the replicating regions after UVC treatment, HeLa cells in early or late S phase (released into S phase for 1.5 and 3.5 h, respectively) were irradiated with 20 J/m$^2$ of UVC, then incubated with culturing medium containing 10uM EdU for 2 h.

Genomic DNA was extracted by PureLink Genomic DNA Mini Kit (Thermo) and sheared by a Q800 Sonicator (Qsonica) to generate fragments averaging 300 bp in length. DNA fragments (1 μg) were used for the Click chemistry reaction. Briefly, the premixed solution A [0.5 mM azide-(PEG)3-biotin (Sigma), 0.1 mM CuSO4 and 0.5 mM THPTA (Sigma) in sodium phosphate buffer (100 mM, pH 7.0)], 5 mM aminoguanidine hydrochloride (TCI, A1129), and 10 mM fresh sodium ascorbate (Sigma) were added to the DNA according to above-mentioned order (all indicated concentrations were final concentrations in the reaction). This reaction was performed at room temperature (RT) for 1 hour with shaking at 1200 rpm. At the end of incubation, 1 mM EDTA was added to terminate the reaction, and the mixture was

purified through a Microspin G50 column (GE Healthcare) which was prewashed with deionized water. The labeled DNA was ethanol precipitated and dissolved in 0.1× TE buffer (1 mM Tris-Cl pH 8.0, 0.1 mM EDTA).

The input libraries were prepared with the NEBNext Ultra II DNA Library Prep Kit (New England Biolabs) according to the manufacturer's instructions, and 1% of products were amplified for input sequencing as described below. The rest of the DNA was purified by streptavidin beads to enrich nascent DNA. Each sample was incubated with 10 µl of pre-blocked Dynabeads MyOne Streptavidin C1 beads (Thermo) in 1× binding buffer (10 mM Tris–Cl pH 8.0, 1 mM EDTA, 0.5 M NaCl, 5 mM MgCl2, 0.1% Tween20, 0.1% CA630) in the presence of 10 µg of salmon sperm DNA at RT for 1 h, then sequentially washed twice with wash buffer I (10 mM Tris–Cl pH 8.0, 1 mM EDTA, 1 M NaCl, 0.1% Tween20, 0.1% CA630), twice with wash buffer II (100 mM Tris–Cl pH 8.0, 1 mM EDTA, 0.5 M LiCl, 1% CA630, 1% sodium deoxycholate), twice with wash buffer III (1 mM Tris–Cl pH 8.0, 0.1 mM EDTA, 100 mM NaCl, 2% SDS), twice with fresh alkaline wash buffer (0.1 M NaOH, 0.1% CA630), once with wash buffer IV (100 mM Tris-Cl pH7.4, 100 mM NaCl, 0.1% CA630), once with 10 mM Tris-Cl pH7.4, and resuspended in 16 µl 0.1× TE. The purified DNA (bound on beads) were amplified with NEBNext Multiplex Oligos for Illumina (New England Biolabs) by NEBNext UltraII Q5 DNA polymerase (New England Biolabs) to generate sequencing libraries which were sequenced in PE150 format on Illumina NovaSeq platform by Mingma Technologies Company.

## Damage-seq and XR-seq libraries preparation and sequencing

Cells were collected in ice-cold PBS at indicated time points and subjected to Damage-seq (HS Damage-seq) and XR-seq as previously described [10, 12]. For Damage-seq, briefly, genomic DNA was extracted with PureLink Genomic DNA Mini Kit and sheared by sonication with a Q800 Sonicator. DNA fragments (1µg) were used for end repair, dA-tailing and ligation with Ad1 by NEBNext UltraII DNA Library Prep kit. Samples were then denatured and subjected to immunoprecipitation with anti-(6–4)PP or anti-CPD antibody (Cosmo Bio), respectively. In order to detect the precise position of lesions, primer Bio3U was attached to purified DNA and extended by NEBNext UltraII Q5 DNA polymerase. After purification, the extension products were annealed to oligo SH for subtractive hybridization. Oligo SH was then removed by incubating with Dynabeads MyOne Streptavidin C1 and the rest of the sample was ligated to Ad2, followed by PCR amplification.

For XR-seq, in brief, fresh cells were lysed by Dounce Homogenizer. After centrifuging to remove chromatin DNA, co-immunoprecipitation with anti-XPG (sc13563, Santa Cruz) antibody was performed to pull down primary excision products generated by nucleotide excision repair. Purified excised fragments were ligated to both 5' and 3' adaptors. Ligation products were further purified by immunoprecipitation with either anti-(6–4)PP or anti-CPD antibody and repaired by corresponding photolyase, followed by PCR amplification and gel purification. Libraries with different indexes were pooled and sequenced in SE50 form on Hiseq 2000/2500 platform by the University of North Carolina High-Throughput Sequencing Facility, or in PE150 form on Hiseq X platform by Mingma Technologies Company. All the sequences of synthesized oligomers used for both assays can be found in references [10, 12].

## Damage-seq sequence pre-analysis

The sequenced reads with adapter sequences (GACTGGTTCCAATTGAAAGTGCTCTTCC-GATCT) at 5' end, were discarded via cutadapt with default parameters for both single-end and paired-end reads [50]. The remaining reads were aligned to the hg19 human genome

using bowtie2 with 4 threads (-p) [51]. For paired-end reads, maximum fragment length (-X) was chosen as 1000. Using samtools, aligned paired-end reads were converted to bam format, sorted using `samtools sort -n` command, and properly mapped reads with a mapping quality greater than 20 were filtered using the command samtools view -q 20 -bf 0x2 in the respective order [52]. Then, resulting bam files were converted into bed format using bedtools `bamtobed -bedpe -mate1` command [53]. Because the exact damage sites should be positioned at two nucleotides upstream of the reads, `bedtools flank` and `slop` commands were used to obtain 10 nucleotide long positions bearing damage sites at the center (5. and 6. positions) [53]. The reads that have the same starting and ending positions, were reduced to a single read by picard tool [54] for deduplication and remaining reads were sorted. Then, reads that did not contain dipyrimidines (TT, TC, CT, CC) at their damage site (5. and 6. positions) were filtered out to eliminate all the reads that do not harbor a UV damage. Lastly, only the reads that were aligned to common chromosomes (chromosome 1–22 + X) were held for further analysis.

## XR-seq sequence pre-analysis

The adaptor sequences (TGGAATTCTCGGGTGCCAAGGAACTCCAGTNNNNNNNAC-GATCTCGTATGCCGTCTTCTGCTTG) at the 3' of the reads were trimmed and sequences without the adaptor sequences were discarded using cutadapt with default parameters [50]. Bowtie2 was used with 4 threads (-p) to align the reads to the hg19 human genome [51]. The aligned single-end reads were directly converted into bam format after the removal of low-quality reads (mapping quality smaller than 20) by samtools [52] and further converted into bed format with `bedtools bamtobed` command [53]. Multiple reads that were aligned to the same position, were reduced to a single read by picard tool [54] to prevent duplication effect and remaining reads were sorted. Like the Damage-seq pre-analysis, only the reads that were aligned to common chromosomes were kept.

## XR-seq and Damage-seq simulation

To simulate damage and repair signals, boquila tool was used by providing input data to it. Boquila is available at: https://github.com/compGenomeLab/boquila. For producing the input data, genomic DNA of asynchronized, early S phased, and late S phased cells were fragmented by sonication, and the libraries were generated by NEBNext Ultra II DNA Library Prep Kit (New England Biolabs) according to the manufactor's instruction. Resulting libraries were sequenced by Illumina Hiseq X platform by Mingma Technologies Company.

## EdU-seq sequence pre-analysis

EdU-seq data was analyzed as it is described in [55]. Briefly, the reads were aligned to hg19 human genome using bowtie2 [51]. Reads with lower quality than 20 and duplicate reads were removed by samtools [52]. Log2 transformed early/late read ratio was calculated in 50 kb long windows. Lastly, the replication domains were generated with a custom R script [55].

## Further analysis

In order to separate a region data (replication domains, initiation zones) into chosen number of windows, the start and end positions of all the regions were set to a desired range with the given command: awk -v a = "$intervalLen" -v b = "$windowNum" -v '{print $1"\t"int(($2+$3)/2-a/2-a*(b-1)/2)"\t"int(($2+$3)/2+a/2+a*(b-1)/2)"\t"$4"\t"""\t"$6}' where "a" is the length of the windows and

"b" is the number of windows. Then, any intersecting regions and regions crossing the borders of its chromosomes were filtered to eliminate the possibility of signal's canceling out effect. After that, `bedtools makewindows` command was used with the `-i srcwinnum` options to create a windowed bed file [53].

To quantify the XR-seq and Damage-seq profiles on the prepared bed file, `bedtools intersect` command was used [53]. Then all windows were aggregated according to their window numbers, and the mean of the total value of each window was calculated. Lastly RPKM normalization was performed, and the plots were produced using ggplot2 in R programming language [56, 57].

### Repair rate and normalized repair rate calculations

Repair rates were calculated separately for each genomic windows by dividing the repair events to damage quantities (XR-seq/Damage-seq). The repair rates were further normalized by the simulated repair and damage signals ($\text{XR-seq}_{real/simulation}$/$\text{Damage-seq}_{real/simulation}$).

### TSS regions

To obtain transcription start sites (TSS), UCSC known canonical genes for hg19 were retrieved from UCSC Table Browser. Using this stranded gene file, TSS regions were created, damage and repair events were mapped on the TSS regions as it is described in further analysis section, and repair rates were calculated. Lastly, transcribed (TS) and non-transcribed (NTS) strands were identified for plotting.

### Generating initiation zones

Generated initiation zones data is publicly available at https://github.com/CL-CHEN-Lab/OK-Seq and can be retrieved from accession no: SRP065949 (NCBI Sequence Read Archive) and PRJEB25180 (European Nucleotide Archive) as raw OK-seq data. To obtain the slope values of initiation zones (not available in the processed files), we downloaded a raw OK-seq data for HeLa cells (SRR code: SRR2913039), GM06990 cells (SRR code: SRR2913063) [36], and IMR90 cells (ENA accession code: ERR2760855) [39]. Then, we trimmed the adaptors with cutadapt [50], aligned to the hg19 human genome with bowtie2 [51], and generated the initiation zones using the script provided for generating initiation zones [36, 58]. For HeLa cells, we separated the generated initiation zones into quartiles based on the adjusted slope values. In addition, we intersected two datasets using `bedtools intersect` command with the `-wa -c -F 0.5` options to obtain initiation zones that are separated into their corresponding replication domains [53]. For all datasets, RPKM of XR-seq was divided to Damage-seq at each window to calculate repair rates after the steps explained in the further analysis section.

### Calculating sequence context around initiation zones

Bed file containing windowed initiation zones converted into fasta using hg19 reference genome. For each window, all possible mono to pentanucleotide sequences were counted, aggregated according to their window numbers and replication domains, and their percentages were calculated with a custom python script. Then, the windows that are at least 2 kb away from the initiation zone center, separated into left and right replicating directions for analyzing sequence asymmetry at the flanking regions of initiation zones. Because we calculated sequence percentages from the forward strand (reference genome), we used its reverse complementary as its percentage on the reverse strand. Therefore, we kept a single sequence from a complementary pair.

### Chromosome-wide replication timing analyses

Replication timing data, Damage-seq, and XR-seq (together with simulated data) were aggregated in 50 kb bins and smoothed using a simple moving average over 10 bins as it was done in [25].

### Identifying A-rich and T-rich sequences

Reference genome was sliced into 50 kb regions and the percentage of As and Ts in forward strand were calculated for each region with a custom python script. Next, T percentage were divided to A percentage, which gave the number of Ts relative to the Ts in the reverse strand. Then, the regions separated into 4 quantiles based on relative Ts and the quantile having the highest relative Ts were tagged as T-rich, while the quantile with the lowest relative Ts (means have higher Ts in the reverse strand) were tagged as A-rich.

### Chromatin states

ChromHMM segmented chromatin states of HeLa cells were retrieved from UCSC [35] and intersected with replication domains as it was done with the initiation zones.

### Genic and intergenic regions

We retrieved UCSC genes from UCSC Table Browser. Then, by subtracting these genes from the genome using the `bedtools subtract` option, we obtained intergenic regions [53]. The code used for calculating the distribution of genic and intergenic regions was adopted from Frigola et al. [59]. In addition, the intergenic regions were also used for extracting initiation zones that are located at the intergenic regions using `intersect` command of bedtools [53].

### Melanoma mutations

Melanoma somatic mutations of 183 tumor samples were obtained from the data portal of International Cancer Genome Consortium (ICGC) as compressed tsv files which is publicly available at https://dcc.icgc.org/releases/release_28/Projects/MELA-AU [60]. Single base substitution mutations were extracted, and only the mutations of common chromosomes were used. To obtain the mutations that could be caused by UV-induced photoproducts, TC>TT and CC>CT mutations were further extracted. Later, mutations were mapped to initiation zones that were separated into their corresponding replication domains. Lastly, mutation counts were normalized by the TC and CC content frequency of the initiation zones.

## Supporting information

**S1 Fig. Flow cytometry analysis.** (A), the distribution of DNA contents of asynchronized HeLa cells (red) was merged with double-thymidine synchronized cells which were released to S phase for 2h (cyan), 4h (blue) and 6h (green). (B), DNA contents of synchronized HeLa cells treated by UVC. Cells were released to S phase for 1.5h (light blue) or 3.5h (dark blue), followed by UV irradiation and 2h-incubation before flow cytometry analysis.
(TIF)

**S2 Fig. Edu-seq data and generated replication domains are similar to the Repli-seq and their replication domains.** (A) Heatmap showing the pairwise comparison of spearman correlation coefficient between UV-treated/untreated early and late S phased EdU-seq samples (2 replicates for each) and G1b, S (1–4), G2 phases of public HeLa-S3 Repli-seq data. (B)

Screenshot of IGV tracks. Tracks 1–6: Repli-seq signals of G1b to G2 phases (purple); track 7: the replication domains determined by Repli-seq (purple); track 8: early (top)/late (bottom) S phase signals of untreated EdU-seq; track 9: the replication domains determined by untreated EdU-seq; track 10: early (top)/late (bottom) S phase signals of UV-treated EdU-seq; track 11: the replication domains determined by UV-treated EdU-seq.
(TIF)

**S3 Fig. Damage and repair maps of UV-irradiated synchronized HeLa cells.** Same as Fig 1 except that synchronized HeLa cells were irradiated at 1.5 h and 3.5 h after release and collected at 2 h after UVC irradiation. (B-C) Replicate A is shown. (D) Replicate A and B are combined.
(TIF)

**S4 Fig. Damage (Damage-seq$_{real/simulation}$) CPDs profiles at 0 minutes and repair (XR-seq$_{real/simulation}$) CPDs profiles at 12 minutes around ERDs (left) and LRDs (right).** Replicate A and B are combined.
(TIF)

**S5 Fig. Chromosome-wide profiles of replication timing and repair.** Replication timing profiles are shown for chromosome 1 and 19 until 50 Mb. Profiles are colored by the local ratio of (A) normalized repair rate and (B) Early/Late S phase normalized repair rate of CPD 12 minutes samples.
(TIF)

**S6 Fig. The effect of chromatin states along with replication timing on repair.** Same as Fig 3 except that the repair of (6–4)PPs was presented. Replicate A and B are combined.
(TIF)

**S7 Fig.** (A) Boxplot of normalized repair rates of CPDs at 12 minutes after UV at early and late S phases for each chromatin state. (B) Relative difference of early S phase and late S phase repair rates ([log2($RR_{Early\ S\ Phase}$/$RR_{Late\ S\ Phase}$)] used for ERDs; [log2($RR_{Late\ S\ Phase}$/$RR_{Early\ S\ Phase}$)] used for LRDs) for each chromatin state. The wilcoxon test was used to assess the significance of differences between ERDs and LRDs. Straight lines were ERDs, and dashed lines were LRDs. Replicate A and B are combined.
(TIF)

**S8 Fig. Log2 transformed odds ratios in each chromatin states.** Chromatin states were categorized as being better repaired in early S phase or late S phase. Then, the association of ERDs (relative to LRDs) and odds of having better repair in early S phase than late S phase was calculated for each chromatin states. Horizontal lines show the lower and upper confidence intervals of odds ratios and Breslow-Day test (with Tarone correction) was used to test homogeneity of odds ratios across the chromatin states.
(TIF)

**S9 Fig. Strand asymmetry of damage and repair around initiation zones in ERDs and LRDs due to sequence context.** (A) Damage and (B) repair profiles of real and simulated reads around initiation zones for CPDs at 0 minutes (for damage), 12 minutes (for repair), and 2 hours (for both damage and repair). Replicate A and B are combined.
(TIF)

**S10 Fig. Strand asymmetry of damage and repair around initiation zones in ERDs and LRDs due to sequence context.** (A) Damage and (B) repair profiles of real and simulated

reads around initiation zones for (6–4)PPs at 0 minutes and 12 minutes, respectively. Replicate A and B are combined.
(TIF)

**S11 Fig. Repair rate asymmetry around initiation zones and replication domains.** (Left) (6–4)PP normalized repair rates [$\log2 (\text{XR-seq}_{\text{real/simulation}}/\text{Damage-seq}_{\text{real/simulation}})$] at 12 minutes in early (A) and late (B) S phases around initiation zones that were separated into corresponding replication domains. (Right) The boxplot of windows in left replicating and right replicating directions are shown separately for plus and minus strands. Paired wilcoxon test was used to assess the significance of differences between the strands. Replicate A and B are combined.
(TIF)

**S12 Fig. Repair rate asymmetry around initiation zones and replication domains in intergenic regions.** Similar as S11 Fig except that damage and repair signals in the annotated transcribed regions were discarded before analysis. Replicate A and B are combined.
(TIF)

**S13 Fig. Repair rate asymmetry around initiation zones and replication domains.** Same as Fig 4C and 4D except that cells were synchronized to the early S phase.
(TIF)

**S14 Fig. Asymmetric repair accumulation around initiation zones in replication domains.** Normalized accumulated repair [$(\text{Damage-seq}_{0h} - \text{Damage-seq}_{2h}) / \text{Damage-seq}_{0h}$] calculated for CPD samples at early and late S phases for ERDs and LRDs.
(TIF)

**S15 Fig. Intergenic repair rate asymmetry around initiation zones in different replication domains.** Similar as Fig 4C and 4D except that damage and repair signals in the annotated transcribed regions were discarded before analysis. Replicate A and B are combined.
(TIF)

**S16 Fig. Repair rate asymmetry around initiation zones and replication domains in intergenic regions.** (A) Distribution of genic and intergenic regions around initiation zones. The blue line indicates genes located on plus strands, the red line indicates genes located on minus strands, and the green line indicates intergenic regions. (B-C) Same as S15 Fig except that cells were synchronized to the early S phase. Replicate A and B are combined.
(TIF)

**S17 Fig. Asymmetric nucleotide content around initiation zones and replication domains.** The percentage of each nucleotide at initiation zones in ERDs (Top) and LRDs (Bottom) was calculated separately for plus (blue) and minus (red) strands. Only a single sequence of a complementary pair was used. Sequences ordered based on the percentage differences between strands (top sequence having the highest difference while the bottom has the least).
(TIF)

**S18 Fig. Asymmetric dinucleotide content around initiation zones and replication domains.** Similar to the S17 Fig except dinucleotides used instead of mononucleotides.
(TIF)

**S19 Fig. Asymmetric trinucleotide content around initiation zones and replication domains.** Similar to the S17 Fig except trinucleotides used instead of mononucleotides. Ten

sequences with the highest difference were shown.
(TIF)

**S20 Fig. Asymmetric quadnucleotide content around initiation zones and replication domains.** Similar to the S17 Fig except quadnucleotides used instead of mononucleotides. Ten sequences with the highest difference were shown.
(TIF)

**S21 Fig. Asymmetric pentanucleotide content around initiation zones and replication domains.** Similar to the S17 Fig except pentanucleotides used instead of mononucleotides. Ten sequences with the highest difference were shown.
(TIF)

**S22 Fig. Correlation of relative occurrence of all mono- to penta-nucleotide sequences on the lagging and leading strand, separated by the number of net TCs (TC-GA).** Same as Fig 4E except net TCs were calculated instead of net Ts.
(TIF)

**S23 Fig. Correlation of relative occurrence of all mono- to penta-nucleotide sequences on the lagging and leading strand, separated by the number of net Cs (C-G).** Same as Fig 4E except net Cs were calculated instead of net Ts.
(TIF)

**S24 Fig. Correlation of relative occurrence of all mono- to penta-nucleotide sequences on the lagging and leading strand, separated by the number of net CCs (CC-GG).** Same as Fig 4E except net CCs were calculated instead of net Ts.
(TIF)

**S25 Fig. Simulation-normalized CPD damage (Damage-seq$_{real/simulation}$) profiles at 0 minutes and repair (XR-seq$_{real/simulation}$) profiles at 12 minutes around initiation zones in ERDs (left) and LRDs (right).** Replicate A and B are combined.
(TIF)

## Author Contributions

**Conceptualization:** Aziz Sancar, Jinchuan Hu, Ogun Adebali.

**Formal analysis:** Cem Azgari.

**Funding acquisition:** Aziz Sancar, Jinchuan Hu, Ogun Adebali.

**Investigation:** Yanchao Huang, Cem Azgari, Mengdie Yin, Yi-Ying Chiou, Laura A. Lindsey-Boltz, Jinchuan Hu.

**Methodology:** Cem Azgari, Jinchuan Hu.

**Project administration:** Jinchuan Hu, Ogun Adebali.

**Supervision:** Jinchuan Hu, Ogun Adebali.

**Visualization:** Yanchao Huang, Cem Azgari.

**Writing – original draft:** Cem Azgari, Jinchuan Hu, Ogun Adebali.

**Writing – review & editing:** Laura A. Lindsey-Boltz, Aziz Sancar, Jinchuan Hu, Ogun Adebali.

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
