## [Decision Letter · Decision Letter 0]

31 May 2022

Dear Dr Adebali,

Thank you very much for submitting your Research Article entitled 'The effects of replication domains on the genome-wide UV-induced DNA damage and repair' to PLOS Genetics.

The manuscript was fully evaluated at the editorial level and by independent peer reviewers. The reviewers appreciated the attention to an important problem, but raised some substantial concerns about the current manuscript. Based on the reviews, we will not be able to accept this version of the manuscript, but we would be willing to review a much-revised version. We cannot, of course, promise publication at that time.

Should you decide to revise the manuscript for further consideration here, your revisions should address the specific points made by each reviewer, which include suggestions on presentation and discussion of your results as well as potential additional analyses and experiments. We will also require a detailed list of your responses to the review comments and a description of the changes you have made in the manuscript. Copy of the revised manuscript with marked up changes would help in re-evaluating of your submission.

If you decide to revise the manuscript for further consideration at PLOS Genetics, please aim to resubmit within the next 60 days, unless it will take extra time to address the concerns of the reviewers, in which case we would appreciate an expected resubmission date by email to plosgenetics@plos.org.  

[LINK]

We are sorry that we cannot be more positive about your manuscript at this stage. Please do not hesitate to contact us if you have any concerns or questions.

Yours sincerely,

Dmitry A. Gordenin, Ph.D.

Associate Editor

PLOS Genetics

Gregory Barsh

Editor-in-Chief

PLOS Genetics

Reviewer's Responses to Questions

**Comments to the Authors:**

Reviewer #1: In this study, Huang et al perform genome-wide measurements of UV damage and NER activity in early S and late S phase (also EdUseq), to examine links between DNA replication program and NER. The central result of the study is summarized in Fig 2B: in late-replicating domains (LRD) there is increased repair rates in the "Late S" experiment compared to "Early S", while this is reversed in early-replicating domains (ERD), suggesting that DNA replication can locally/regionally promote activity of NER, speculatively via chromatin relaxation.

Major:

1- Repair rate is a composite of two measurements: damage and repair activity. Can they compare each of the two, between "Early S" and "Late S" experiments, in ERD and in LRD. We would need to see that it is repair (XR-Seq) that is changing not the damage. Otherwise I think the data is not compatible with the model "chromatin opens up (due to replication) to facilitate repair".

2- "indicating that repair rate was elevated when the region started to be replicated. This effect is likely caused by the relaxing of local chromatin during replication (33-35)." -- I think "likely" probably doesn't cut it if they wish to report this relaxing mechanism as a major finding in the Abstract or the Author Summary. Chromatin accessibility was not directly measured, unfortunately, so what remains is the proxy analysis that stratifies the data by chromatin state (Fig. 3). The results here seem unclear in terms of supporting (or not) the model. The major (by genomic coverage) active states "transcription-associated" and "low activity proximal" seem to show the same direction of effect as the major inactive states "polycomb repressed" and "HET/Rep". I am not sure this supports the chromatin opening hypothesis (the difference, though, between the active promoters and inactive promoters is notable and might support this). I think a possibly more illuminating analysis would be checking ERD vs LRD effects (like Fig 2a/2b) within each state, and/or vice versa -- stratifying for one variable while checking another.

3- Regarding the strand-assymetry in the repair process, which they ascribe to "intrinsic sequence features", here referring to extended sequence motifs of poly-T. This analysis, while interesting, seems a bit preliminary.

3a. Could they systematically check various types of sequence motifs not only polyT, to show that polyT is indeed special (if it is).

3b. Importantly, if the sequence motif matters for the strand-assymetry of repair rate, then this effect should be independant of the DNA replication program and should be equally observed in any location of the genome independently of the replication initiation zones.

Minor:

- Title of Fig. 1 "Ongoing replication fork stimulates nearby nucleotide excision repair" does not appear to match its contents. Fig 2 has no title.

- The dip in repair rates at the TES (Fig 1D) - is this known, if not could they comment on it?

- Also, was the repair-rate in Fig 1D around TSS/TES normalized by the simulated repair rates (controlling for sequence composition)

- Reference 36 (Zheng et al.) should probably be in the introduction already. It provides important prior context for the here-reported finding of increased repair rates in early-replicating (euchromatic) domains compared with late-replicating (heterochromatic) ones.

- Fig 2b typo "polycob"

- Line 178 starting with "Thus," on a first reading it was not immediately clear to me why there was a need for the additional 2h experimental timepoint. They could make their reasoning more explicit.

- Lines 159-209 section was a bit technical and difficult to parse. They could consider shortening and/or rewriting for clarity.

Reviewer #2: The authors attempted to build relationships between DNA damage and repair processes after UV irradiation of HeLa cells at different phases of the cell cycle and the parameters of DNA replication, including early and late replication and strand asymmetry. As shown before, early replicating regions are repaired faster than late replicating regions presumably due to open chromatin structure and active genes in those parts of the genome. They concluded that UV lesions on the lagging strand were repaired differently in late replication domains, which was probably due to an imbalanced sequence context with more T-rich sequences on lagging strands favoring cyclobutane dimer (CPD) formation. This paper contains few new observations and is presented with many conceptual and methodological problems. I also question the novelty of this paper in terms of leading versus lagging strand data when considering reference 27.

1) They used two methods to assess DNA damage: damage-seq and XR-seq. XR-seq measures a combination of initial damage levels and repair efficiency in the form of in vivo excised short oligonucleotide fragments that are sequenced. Because these are two very different methods, it seems questionable to specifically infer DNA repair rates by normalizing XR-seq versus damage-seq data. A more straightforward approach would be to perform damage-seq at various time points.

2) When looking at XR-seq data from a 12-minute time point after irradiation, this is likely measuring only very fast repair kinetics and should reflect primarily (6-4) photolesions. The authors are not measuring real repair rates with XR-seq other than a snapshot of repair, and only two timepoints were used (12 min and 2 hours).

3) Figure 1D:

The strong DNA repair peak at the TSS contrasts entirely with a dominant mutation peak in melanoma at the TSS as published by Perera et al (Nature, 2016).

4) The authors repeatedly work with simulated damage and repair rates but never explain how they measured these. What is the purpose of using “simulated” data rather than real data?

5) On line 184, the authors state that transcription-coupled repair might play a role at 2h but not at 12 min. That makes little sense to me.

6) Some sections of the paper are confusing. See lines 188/189 and lines 206-208. Part of the ambiguity is probably caused by the conflation of DNA repair and DNA damage levels within XR-seq data.

7) Figure 4:

The repair data for panels B and C look different even though they both are from the same time point.

What do the authors mean by left replicating and right-replicating?

Panel E. The number of dipyrimidines is more important than the net numbers of T-A.

8) The authors used replication data from HeLa cells and from two fibroblast cell lines for comparisons with mutation data from melanomas. However, how much are the replication patterns cell type-dependent? According to Figure 5, the number of shared initiation zones is quite limited between the three cell lines, and HeLa is the greatest outlier. For these reasons it seems inappropriate to extrapolate from these cell line replication data to mutations in melanocytic tumors.

9) They used non-physiological UVC radiation. Most comparisons from such data cannot be related to human skin cancer mutations.

10) Methods: What does it mean that replicates A and B were combined throughout the analysis?

Abstract: It is speculative to say that replication contributes to accumulating T-tracts and high AT content on lagging strands of initiation zones during evolution. There is no data to support such an assumption.

Reviewer #3: In “The effects of replication domains on the genome-wide UV-induced DNA damage and repair,” Huang et al., investigate the impact of DNA replication of UV damage repair efficiency. It was previously known that UV-induced mutations in cancers are increased while nucleotide excision repair (NER) is decreased in late replicating, lowly expressed heterochromatic domains of the human genome. However, the relative contribution of transcription and replication to opening chromatin to allow NER to function in heterochromatic domains is unknown. The authors therefore utilize Edu-seq to map early and late replicating domains in HeLa cells and then independently map CPDs and 6-4 photoproducts using damage-seq and XR-seq methods in synchronized cells in either the early or late phase of DNA replication. The authors find that NER efficiency in both the early and late replicating domains is elevated when each domain is being actively replicated, suggesting that DNA replication plays some role in providing access of NER to UV induced lesions. This effect is most dramatic for CPDs over 6-4 photoproducts. Effects of active replication on NER repair occur in multiple chromatin domains across these regions. Damage-seq, XR-seq reads, and UV signature mutations in melanoma also occurred more readily in replication initiation zones on the lagging strand, consistent with previous results (2). However, T:A richness of the lagging strand was identified and would provide more potential sites of UV lesions to be formed on the lagging strand, suggesting some of the lagging strand bias may be due to sequence context. Overall, the manuscript is fairly well written and easy to follow. The determination that replication increases NER efficiency is a significant discovery towards our understanding of how mutational heterogeneity is established in human cancer cells. I believe addressing the following specific points will strengthen the manuscript.

Specific points

1. In line 113, the authors use the lack of transcriptional strand bias at 12 min post-exposure to argue that their analysis on replication does not have the exclude the impacts of transcription. However, transcription is still proceeding during these 12 minutes and can also impact global genomic NER, especially in heterochromatic regions (3). Because of the strong association of replication timing with transcriptional activity, it is difficult to separate the contribution of each of these processes on chromatin state through mapping techniques. Additional discussion of this point or measuring damage-seq and XR-seq in the presence of a transcription inhibitor like actinomycin D would solidify this statement.

2. The effect size of DNA replication on promoting NER appears to be relatively small. In Fig. 2B its clear that despite ongoing replication, NER in late replicating regions is still greatly inhibited compared to in early replicating regions. Is this difference do to the higher transcriptional activity of the early replicating domains? Stating specific fold changes or odds-ratios throughout the text would help establish how much of an impact replication is having on NER rate as would a discussion of how these effect sizes compare to other influences on NER.

3. In line 148, the authors state that the difference between repair efficiency with and without replication is minor for more efficiently repaired regions. However, in fig. 3 they show that inactive promoters, CTCF insulators, and Low activity proximal states are three of the four non-significant chromatin states and they have middle to low repair rates indicated in Fig. 3A. This seems to be an over-simplification of the data.

4. The authors interpret the lagging strand bias of UV damage, NER, and mutation to be caused by an underlying sequence bias for T in the lagging strand which would provide more dipyrimidines for damage on the lagging strand. However, the biases seen for UV damage and NER is greater than the simulated bias based on sequence content alone. Additionally, the mutation analysis in Fig. 5 standardizes by sequence content. This suggests that other factors also influence the increased lesions on the lagging strand. What some of these possibilities are should be discussed.

5. The use of OK-seq defined initiation zones seems to limit the focus of the analysis to specific regions of the genome. Are similar effects seen if overall replication direction assessments (like those is (1), are used?

6. While generally well-written, there are small typos throughout the manuscript that should be corrected. An example is in the title, which should probably be: The effects of replication domains on genome-wide UV-induced DNA damage and repair. The inclusion of the word “the” doesn’t make sense.

References:

1. Haradhvala NJ, Polak P, Stojanov P, Covington KR, Shinbrot E, et al. 2016. Mutational Strand Asymmetries in Cancer Genomes Reveal Mechanisms of DNA Damage and Repair. Cell 164:538-49

2. Seplyarskiy VB, Akkuratov EE, Akkuratova N, Andrianova MA, Nikolaev SI, et al. 2019. Error-prone bypass of DNA lesions during lagging-strand replication is a common source of germline and cancer mutations. Nat Genet 51:36-41

3. Zheng CL, Wang NJ, Chung J, Moslehi H, Sanborn JZ, et al. 2014. Transcription restores DNA repair to heterochromatin, determining regional mutation rates in cancer genomes. Cell Rep 9:1228-34

**Have all data underlying the figures and results presented in the manuscript been provided?**

Reviewer #1: Yes

Reviewer #2: Yes

Reviewer #3: Yes

PLOS authors have the option to publish the peer review history of their article (what does this mean?). If published, this will include your full peer review and any attached files.

Reviewer #1: No

Reviewer #2: No

Reviewer #3: No

---

## [Decision Letter · Decision Letter 1]

11 Aug 2022

Dear Dr Adebali,

Thank you very much for submitting your Research Article entitled 'Effects of replication domains on genome-wide UV-induced DNA damage and repair' to PLOS Genetics.

The manuscript was fully evaluated at the editorial level and by independent peer reviewers. The reviewers appreciated the attention to an important topic but identified some concerns that we ask you address in a revised manuscript.

We therefore ask you to modify the manuscript according to all reviewers recommendations. Your revisions should address the specific points made by each reviewer.

Please, note that reviewer 3 recommended evaluation of statistical significance of differences between odds ratios to support apparent differences for values and for directions of repair efficiencies.  For this purpose, you may find useful the Breslow-Day Test for Homogeneity of the Odds Ratios (not very often used in genomics date analyses).  This reviewer also repeatedly pointed on the lack of numeric source data used for building graphs or for generating summary statistics in your manuscript .  As outlined below, this is a requirement of our data availability policy   

[LINK]

Yours sincerely,

Dmitry A. Gordenin, Ph.D.

Academic Editor

PLOS Genetics

Gregory Barsh

Editor-in-Chief

PLOS Genetics

Reviewer's Responses to Questions

**Comments to the Authors:**

Reviewer #1: In the revised manuscript, Huang et al performed some additional analysis in response to my queries. I have only minor suggestions remaining. Generally I think that the study and presentation has been improved by the revisions.

There is added value in the analyses that clarified the role of various types/lengths of repeats. They should consider plotting some of this data as an additional main figure rather than keeping it buried in Fig S20-24.

Regarding the role of chromatin opening mediating the mechanism they report, we now have it clear in the text their study does not in fact have data to support this "This effect could be, at least partially, attributed to the relaxing of local chromatin during replication (34-36)." I think they should also correct the Abstract by removing "by relaxing surrounding chromatin".

I think their Fig R1A is an interesting visualization because it supports tht the more likely causal role in shaping repair rates lies with replication time, rather than with eu/heterochromatin, if I am reading this correctly. I would suggest to include this figure in the manuscript as it provides a different, more informative emphasis than Fig 3 currently provides.

Reviewer #2: I am overall satisfied with the authors' responses.

Reviewer #3: Dr. Ogun Adebali and colleagues have submitted their revision of “Effects of replication domains on genome-wide UV-induced DNA damage and repair.” In general, the authors have addressed my previous comments. However, I have some minor suggestions for clarification to the text.

1) When addressing the differential impact of replication to chromatin compaction on NER repair rates, the authors conclude in the rebuttal that “genome-wide repair profile mainly depends on chromatin compaction and transcription activity, while replication exerts an additional impact.” I did not see this statement added to the manuscript. I think it should be included somewhere.

2) The lack of significance for differences in repair efficiency while replicating or not in inactive promoters and CTCF insulations are clearly due to low numbers, which the authors now provide. This leaves only Low activity proximal sites as non-significant. However, the numbers provided suggest that repressed polycomb and heterochromatin are significant, but in the opposite direction. This would fit the authors’ interpretation. It may be helpful to provide an odds-ratio to help emphasize this, instead of just providing the statistical significance.

3) I am unclear how Figure S5 contains information about repair strand symmetry. Please clarify this.

**Have all data underlying the figures and results presented in the manuscript been provided?**

Reviewer #1: Yes

Reviewer #2: Yes

Reviewer #3: **No: **There does not appear to be an numerical data for graphs or summary statistics in spreadsheet form.

PLOS authors have the option to publish the peer review history of their article (what does this mean?). If published, this will include your full peer review and any attached files.

Reviewer #1: No

Reviewer #2: No

Reviewer #3: No

---

## [Decision Letter · Decision Letter 2]

12 Sep 2022

Dear Dr Adebali,

We are pleased to inform you that your manuscript entitled "Effects of replication domains on genome-wide UV-induced DNA damage and repair" has been editorially accepted for publication in PLOS Genetics. Congratulations!

Yours sincerely,

Dmitry A. Gordenin, Ph.D.

Academic Editor

PLOS Genetics

Gregory Barsh

Editor-in-Chief

PLOS Genetics

Comments from the reviewers (if applicable):

Reviewer's Responses to Questions

**Comments to the Authors:**

Reviewer #1: The authors have satisfactorily addressed my queries. In my opinion the manuscript would be ready for publication.

Reviewer #3: The authors have addressed all of my previous comments.

**Have all data underlying the figures and results presented in the manuscript been provided?**

Reviewer #1: Yes

Reviewer #3: Yes

PLOS authors have the option to publish the peer review history of their article (what does this mean?). If published, this will include your full peer review and any attached files.

Reviewer #1: No

Reviewer #3: No

**Data Deposition**

http://datadryad.org/submit?journalID=pgenetics&manu=PGENETICS-D-22-00489R2

**Press Queries**

---

## [Editor Report · Acceptance letter]

21 Sep 2022

PGENETICS-D-22-00489R2 

Effects of replication domains on genome-wide UV-induced DNA damage and repair 

Dear Dr Adebali, 

We are pleased to inform you that your manuscript entitled "Effects of replication domains on genome-wide UV-induced DNA damage and repair" has been formally accepted for publication in PLOS Genetics! Your manuscript is now with our production department and you will be notified of the publication date in due course.

With kind regards,

Olena Szabo

PLOS Genetics

On behalf of:
